# A Bitter Taste Receptor as a Novel Molecular Target on Cancer-Associated Fibroblasts in Pancreatic Ductal Adenocarcinoma

**DOI:** 10.3390/ph16030389

**Published:** 2023-03-03

**Authors:** Jessica Hung, Samantha M. Perez, Siva Sai Krishna Dasa, Sarah P. Hall, Danielle B. Heckert, Brian P. Murphy, Howard C. Crawford, Kimberly A. Kelly, Lindsey T. Brinton

**Affiliations:** 1Department of Biomedical Engineering, University of Virginia, Charlottesville, VA 22908, USA; 2ZielBio Inc., Charlottesville, VA 22902, USA; 3Department of Molecular and Integrative Physiology, University of Michigan, Ann Arbor, MI 48109, USA; 4Rogel Comprehensive Cancer Center, University of Michigan, Ann Arbor, MI 48109, USA; 5Henry Ford Pancreatic Cancer Center, Henry Ford Health, Detroit, MI 48202, USA

**Keywords:** TAS2R9, CXCR2 inhibitor, targeted liposome, stromal therapy

## Abstract

Cancer-associated fibroblasts (CAFs) execute diverse and complex functions in cancer progression. While reprogramming the crosstalk between CAFs and cancer epithelial cells is a promising avenue to evade the adverse effects of stromal depletion, drugs are limited by their suboptimal pharmacokinetics and off-target effects. Thus, there is a need to elucidate CAF-selective cell surface markers that can improve drug delivery and efficacy. Here, functional proteomic pulldown with mass spectrometry was used to identify taste receptor type 2 member 9 (TAS2R9) as a CAF target. TAS2R9 target characterization included binding assays, immunofluorescence, flow cytometry, and database mining. Liposomes conjugated to a TAS2R9-specific peptide were generated, characterized, and compared to naked liposomes in a murine pancreatic xenograft model. Proof-of-concept drug delivery experiments demonstrate that TAS2R9-targeted liposomes bind with high specificity to TAS2R9 recombinant protein and exhibit stromal colocalization in a pancreatic cancer xenograft model. Furthermore, the delivery of a CXCR2 inhibitor by TAS2R9-targeted liposomes significantly reduced cancer cell proliferation and constrained tumor growth through the inhibition of the CXCL-CXCR2 axis. Taken together, TAS2R9 is a novel cell-surface CAF-selective target that can be leveraged to facilitate small-molecule drug delivery to CAFs, paving the way for new stromal therapies.

## 1. Introduction

Pancreatic ductal adenocarcinoma (PDAC) is the third leading cause of cancer-related deaths in the United States, with a 5-year survival rate of only 11% [1,2]. The poor clinical outcomes reflect the lack of effective treatments and ineffective strategies to mitigate the complex involvement of the stromal compartment. In PDAC, in particular, extensive desmoplastic stroma occupies >70% of total tumor volume [3,4,5]. The most abundant cellular components in the stroma are cancer-associated fibroblasts (CAFs), which form complex networks with tumor epithelial cells, immune cells, and endothelial cells. CAFs also play an integral role in regulating acellular components (collagen, hyaluronan, chemokines, and cytokines), tumor progression, immunosuppression, metastasis, and drug resistance [6,7,8,9]. Emerging evidence supporting the importance of the stroma in PDAC led to the therapeutic approaches that deplete the stroma. Clinically (NCT01130142, NCT01959139), anti-CAF drugs reduced the stroma, but unexpectedly resulted in a more aggressive tumor phenotype, enhancement of tumor immune evasion, and development of chemoresistance [10,11]. These results emphasize the complex role of the stroma in balancing tumor-promoting and -restraining functions [12,13] and the necessity of non-cytotoxic, CAF-directed approaches. 

In contrast to depletion, reprogramming the crosstalk between the CAFs and the tumor microenvironment offers an opportunity to tilt CAF functions in favor of its tumor-restraining roles. In this way, CAF-targeted therapies could work with existing regimens to alleviate stromal effects that counter drug efficacy. In one notable example, Sano et al. showed that cancer cells and CAFs promoted one another’s invasion and migration through the CXC chemokines-receptor (CXCLs-CXCR2) axis; blocking the CXCL-CXCR2 axis inhibited PDAC microinvasion and prolonged survival [14]. In another study, Biffi et al. demonstrated that a JAK inhibitor shifted CAFs from an inflammatory to a myofibroblastic phenotype, ultimately decreasing tumor growth [15]. Taken together, these studies highlight the essential functions of CAFs in tumor progression and underline the therapeutic potential of strategies reprogramming CAF activity.

As conceptual proof, we sought to leverage a CAF-selective peptide, HTTIPKV, that we previously developed [16] to create a CAF-selective drug delivery system. The HTTIPKV peptide exhibited a sequence unique to CAFs when queried against our peptide signature database, which at the time spanned 56 cell lines and tissues and was selective for CAFs across in vitro and in vivo experiments. As such, HTTIPKV presented a unique opportunity to develop a CAF-targeted drug delivery system. However, we had yet to uncover the binding partner of the HTTIPKV peptide. We now present taste receptor type 2 member 9 (TAS2R9) as a previously unknown CAF-selective marker. TAS2R9 is a G protein-coupled receptor (GPCR) that recognizes bitter stimuli in the oral cavity, but is also expressed in the airway epithelium and the gastrointestinal tract [17,18]. Herein, we report and validate TAS2R9 expression in pancreatic CAFs and demonstrate that TAS2R9 is a viable molecular target for stroma-directed therapy.

Next, we sought to demonstrate how TAS2R9-targeted liposomes could modulate CAF function. We encapsulated the CXCR2 inhibitor, SB225002, in a liposome and conjugated HTTIPKV peptides to the surface. CXCR2 is a GPCR for CXC chemokines, CXCL1-3, and CXCL5-8, and is involved in inflammation, angiogenesis, tumorigenesis, and metastasis through tumor-stroma interaction [14,19,20,21,22,23]. Pharmacological inhibition of CXCR2 improved T-cell infiltration in a pancreatic syngeneic murine model and enhanced sensitivity to anti-PD1 immunotherapy [21]. Despite these promising features, the safety of systemic CXCR2 inhibitors remains questionable as these small molecule drugs disrupt neutrophil chemotaxis and activation, resulting in an increased risk of developing neutropenia in cancer patients who already suffer from a compromised immune system [24,25]. Furthermore, we questioned if targeted liposomal drug delivery of a small molecule to its target cell type could improve efficacy. Thus, we sought to evaluate the liposomal pharmacokinetics and the pharmacological responses to SBS225002 when delivered to CAFs. We observed selective accumulation of the targeted liposomes in PDAC stroma in murine xenograft models. Compared to non-targeted liposomes, TAS2R9-targeted liposomes constrained tumor growth by about 50%. The discovery of TAS2R9 as a CAF-selective marker whose targeting can improve the anti-tumor effects of liposomal delivery of a CXCR2 inhibitor opens the door for the modulation of CAF activity as a potent therapeutic strategy in PDAC.

## 2. Results

### 2.1. Identification of TAS2R9 as New CAF-Selective Target

Previously, we identified two 7-amino acid peptides, HTTIPKV and APPIMSV, as having high specificity and selectivity for CAFs isolated from the stroma of PDAC patients [16]. Thus, we hypothesized that their binding partners could serve as potential cellular markers and therapeutic targets for CAFs. Using a phage display-based functional proteomics approach, wild-type (WT) M13KE phage or phage displaying CAF selective peptides (phHTTIPKV and phAPPIMSV) were incubated with PDAC patient-derived CAFs allowing the phage to serve as “bait.” Phage were chemically modified to crosslink to their cell surface binding partner, then phage pull-down samples were resolved via SDS-PAGE. This revealed the presence of a unique band in the HTTIPKV sample when compared with the WT M13KE phage and phAPPIMSV (Figure 1A), which was excised, digested with trypsin, and analyzed via mass spectrometry. Two unique tryptic digest fragments were identified, matching 12% coverage to human TAS2R9, thus revealing TAS2R9 as a candidate target (Appendix A). In-proteo ELISA experiments were performed to evaluate the specific binding of phHTTIPKV to recombinant human TAS2R9 (rhTAS2R9). Phage displaying KTLLPTP (phKTLLPTP), a PDAC epithelial cell-specific peptide [26], served as a negative control. To account for the background binding from the phage itself, phHTTIPKV’s and phKTLLPTP’s binding to rhTAS2R9 was normalized to that of M13KE (no peptide displayed). A 2.72-fold increase in specific binding to rhTAS2R9 was observed from phHTTIPKV compared to phKTLLPTP (Figure 1B), indicating selectivity for TAS2R9, which is comparable to other successful peptide-targets [26]. Therefore, we pursued validation of TAS2R9 as HTTIPKV’s target and as a novel CAF-selective cell surface marker.

### 2.2. Validation of TAS2R9 as a CAF Target

To validate TAS2R9 as a CAF target, we first quantified its expression in CAFs at the mRNA (qPCR) and protein (flow cytometry and IF) level. qPCR confirmed the presence of TAS2R9 RNA transcripts in CAFs (Appendix A). Thus, we evaluated TAS2R9 mRNA expression in normal and malignant fibroblasts and showed that TAS2R9 had a 57-fold higher mRNA expression in CAFs compared to human dermal fibroblasts (HDFs) (Figure 1C, *p* = 0.009). Further investigation of TAS2R9 as a target demonstrated the presence of TAS2R9 on a large population of the CAF cell line. Flow cytometry showed a distinct shift compared to secondary only or vehicle, indicating the presence of TAS2R9 on the majority of these cells (Figure 1D). Immunofluorescence showed a TAS2R9 staining pattern (green) consistent with both cytoplasmic and membrane-associated TAS2R9 (Figure 1E).

### 2.3. TAS2R9 mRNA Expression Is a Prognostic Indicator in Several Cancers

Leveraging publicly available databases, we surveyed the status of TAS2R9 mutations and mRNA expression in human cancers. Analysis of 10,953 patient samples across 32 cancer types from the TCGA pan-cancer atlas database revealed that TAS2R9 was altered in 1.9% (211/10,953) of samples with a somatic mutation frequency of 0.5%. The 65 mutations identified consisted of missense and truncating passenger mutations that spanned the whole gene, with the highest mutations occurring at the R213*/Q site (Figure 2A). Only one case of a missense mutation was reported in queried PDAC samples. A review of TAS2R9 alteration frequencies demonstrates that amplification is the most prevalent alteration, occurring in 2% (3/150) of PDAC samples (Figure 2B). Overall, TAS2R9′s sporadic and low-frequency mutations suggest this is not a major driver of TAS2R9′s role in carcinogenesis. 

Analysis of TAS2R9 gene expression revealed elevated mRNA expression in 7.33% (11/150) of curated PDAC patients from the TCGA pan-cancer atlas database (Appendix A) [27]. Furthermore, an mRNA co-expression analysis of TAS2R9 with canonical gene markers of CAF subtypes revealed a weak Spearman’s correlation with CD74 (ρ = 0.143; *p* = 0.0818) and HLA-DRA (ρ = 0.108; *p* = 0.189), which are makers for antigen-presenting CAFs (apCAF, Figure 2C) [28]. PDAC tumors with high TAS2R9 expression showed low-to-no expression of APOD, POSTN, and PLA2G2A, gene markers corresponding to inflammatory CAF (iCAF), myofibroblastic CAF (myCAF), and metabolic CAF (meCAF), respectively (Figure 2C). This expression profile matches the features of a previously reported CAF subpopulation with weak antigen-presenting function but distinct characterization from the apCAF, iCAF, myCAF, and meCAF populations [28]. These findings support the exciting investigation of TAS2R9 expression as independent of APOD, POSTN, and PLA2G2A and as a novel marker for a distinct CAF subtype.

An in-silico prediction of TAS2R9 protein interactors identified several taste receptor family members (TAS1R1, TAS1R2, and TAS1R1) and glutamate receptors (GRM2, GRM4, GRM6, GRM7, and GRM8) as functional partners (Figure 2D). This network is consistent with TAS1R and TAS2R receptors’ canonical role in mediating response to sweet or bitter tasting stimuli, respectively [29,30]. Beyond its role in the gustatory system, TAS2R9 expression also serves as an indicator of patient prognosis across multiple cancer types (Figure 2E,F). While a patient survival analysis of TAS2R9 mRNA expression in PDAC patients (*n* = 150) revealed it was not significantly associated with worse overall survival (OS), higher TAS2R9 expression was an indicator of worse OS in several cancers with dense stroma including breast, ovarian, and head and neck squamous cell carcinoma [31,32,33]. These results suggest that TAS2R9′s role in cancer could differ depending on the tissue type or tumor microenvironment composition. Thus, an expanded analysis into how TAS2R9′s protein interaction network differs in tumors could help explain its role in tumorigenesis and the prognostic value of its mRNA expression across multiple cancer types.

### 2.4. Development of a TAS2R9-Targeted Drug Delivery System

Building off of work demonstrating superior pharmacodynamics and efficacy of liposomes with targeting peptides [39,40,41], we sought to create a CAF-selective liposome that would limit the drug effect to the stromal compartment. Leveraging the selectivity of TAS2R9 for CAFs and the specificity of the HTTIPKV peptide for TAS2R9, we engineered TAS2R9-targeted liposomes through conjugation of HTTIPKV to PEG-lipid moieties present in the liposome (Figure 3A). The peptide sequence HTTIPKVGGSK(fitc)C was conjugated to DSPE-PEG3400-maleimide to form peptide-DSPE-PEG (Figure 3A). Liposomes without surface modifications were prepared in parallel as a negative control liposome. A non-exchangeable lipid dye, DiR, was incorporated into the lipid formula to allow particle tracking via imaging. All batches of targeted and control liposomal formulations were of similar particle size (100–120 nm, Appendix A) and concentration (2.4 × 10^12^ – 3.2 × 10^12^ particles/mL) as determined by nanoparticle tracking analysis on a NanoSight Instrument (Figure 3B). The zeta potential of control and targeted liposomes was −37.2 mV and −34.2 mV, respectively, indicating that the peptide did not alter the charge of the liposomes (Appendix A).

We measured the binding kinetics of the targeted and control liposomes by Octet. His- rhTAS2R9 was non-covalently bound to the biosensor then the biosensor was exposed to 40 μM of either control or targeted liposomes. An association with TAS2R9 was observed at 400 sec when exposed to targeted liposomes, meanwhile, no association was seen when TAS2R9 was exposed to the control liposomes, indicating specific binding of targeted liposomes to TAS2R9.

### 2.5. TAS2R9-Targeted Delivery Increases Liposome Accumulation in Tumor of Admix PDAC Model

We next evaluated the in vivo capacity of the TAS2R9-targeted liposomes to reach PDAC stroma. We intravenously injected targeted and control liposomes via the tail vein in a subcutaneous xenograft admixture PDAC mouse model containing both BXPC3 and CAF cells (BXPC3-to-CAF-ratio = 1:3) as previously described [16]. The ratio of CAF to cancer cells was chosen to reflect the dense stromal content in PDAC tumors in patients. Since the liposomes contained a non-exchangeable lipophilic dye (DiR), we could detect the location and amount of accumulation non-invasively using fluorescent molecular tomography (FMT) imaging (Figure 4A). The amount of DiR in the tumor xenograft was quantified from the reconstructed images using the TrueQuant (FMT system) software (Figure 4B). We used a compartment model to fit the liposome time course data and calculate area under the curve (AUC). A 1.9-fold higher liposome accumulation was observed in targeted over control liposomes, indicating an increase in total drug exposure (Figure 4C,D; AUC 13.89 in targeted versus 7.19 in control liposomes). This amount of increased accumulation is comparable with published targeted drug delivery systems in vivo where a two-fold increase in drug delivery improved therapeutic outcomes [42,43]. 

In order to assess the distribution of the liposomes within the tumor, tumors from an additional study with mice harboring admix BXPC3/CAF tumors were harvested 24 h post liposomal injection, embedded in optimal cutting temperature (OCT) compound, cryosectioned, and imaged via confocal microscopy. We co-stained for anti-α-SMA, a histopathological marker of the stromal regions, to assess localization of targeted liposomes with the stromal compartment tumor sections. Targeted liposomes showed a higher degree of overlap with stromal regions than control liposomes (Mander’s correlation coefficient = 0.22 for targeted vs 0.045 for control liposomes; *p*-value = 0.0002), indicating that liposomes appeared to bind to CAFs (Figure 4E,F). Compared to control liposomes that are found non-specifically throughout the tumor sections, the incorporation of TAS2R9-targeting peptides shifted the liposomal distribution, favoring PDAC stroma (Figure 4F). 

### 2.6. TAS2R9-Targeted Liposomal Delivery of a CXCR2 Inhibitor Inhibits Tumor Growth

Inhibiting the CXCL-CXCR2 axis with CXCR2 antagonists has been shown to inhibit tumor growth, extend survival, and induce anti-angiogenesis effects in tumor xenograft models [20,21]. However, systemic administration of CXCR2 inhibitors increases the risk of developing neutropenia [24,25], potentially leading to adverse effects in cancer patients with compromised immune systems. Thus, we investigated if the delivery of a CXCR2 inhibitor by TAS2R9-targeted liposomes resulted in enhanced therapeutic efficacy. We developed a liposomal encapsulated formulation of a CXCR2 inhibitor, SB225002, using the thin-film hydration method. Fresh batches of liposomes were produced weekly to circumvent potential drug leakiness from the liposomes. We compared tumor outgrowth in mice bearing subcutaneous admixture xenografts of both BXPC3 and CAF during four treatment regiments over the course of 19 days: (1) untreated, (2) free drug (0.5 mg/kg, i.p., 5x/week), (3) drug-loaded no-peptide liposomes (0.83 mg/kg, i.v., 3x/week), and (4) drug-loaded targeted liposome (0.83 mg/kg, i.v., 3x/week). The free SB225002 was administered based on previously described optimal dosage and route [20]. The liposomal drug dosage was calculated for each injection to achieve the same total weekly drug administration for all groups.

Mice from all groups treated with SB225002 displayed significant inhibition of tumor growth compared to the untreated mice, with the greatest inhibition seen in the CAF-targeted liposome-treated cohort (Figure 5A). Unexpectedly, we observed tumor ulceration in the control liposome cohort on day 16 and had to euthanize the animals; no ulceration was observed in any of the other groups.

Compared to systemic delivery, SB225002 delivered by CAF-targeted liposomes resulted in a respective 1.6- and 1.25-fold smaller tumors on day 12 and 16 (*p* < 0.05), indicating improved anti-tumor efficacy with CAF-targeted therapies (Figure 5B). There was no statistically significant difference between targeted liposome and free drug treatments on day 19. At the end of treatment (day 19), the average tumor volume of mice receiving SB225002-loaded targeted liposomes was significantly smaller compared to the untreated mice (Figure 5B; 1.9-fold smaller, *p* < 0.05). H&E staining of tumor sections revealed a similar tumor-to-stromal ratio from all groups, indicating that the decrease in tumor volume was not merely a reflection of stromal depletion (Appendix A). Taken together, these results demonstrate that TAS2R9-targeted delivery of SB225002 to PDAC stroma results in enhanced inhibition of tumor growth compared to free drug and untargeted liposome delivery.

### 2.7. Drug-Loaded TAS2R9-Targeted Liposomes Show Target Pathway Engagement and Decrease Cancer Cell Proliferation

To further evaluate pharmacodynamics, we characterized CXCR2 inhibition as a surrogate for the delivery of SB225002. Tumors from all groups were fixed and paraffin embedded for immunohistochemistry staining. Anti-CXCR2 staining revealed a 50% and 45% decrease in CXCR2 protein expression in CAF-targeted liposome-treated tumors compared to tumors from free-drug or untargeted liposome treatment, respectively (*p* < 0.05, Figure 5E,F). These data suggested an effective delivery of SB225002 by targeted delivery. Furthermore, the inhibition of angiogenesis has been cited as a contributing factor to the anti-tumor effects of SB225002 [20]. Thus, we evaluated if the targeted and free drug delivery of SB225002 disrupted angiogenesis in vivo. We stained tumor sections with CD31 and quantified vessel volume fractions using ImageJ. No significant difference was observed between the four treatment groups (Figure 5B,C). Next, we examined SB225002’s effects on cell proliferation through Ki67 staining on the tumor sections. Strikingly, mice treated with free drug and drug-loaded targeted liposomes had a significant reduction in Ki67+ area fraction with no decrease in proliferation observed from untargeted liposomal treatment (*p* < 0.01, Figure 5C). Overall, we demonstrated that treatment with TAS2R9-targeting liposomes loaded with a CXCR2 inhibitor results in decreased cancer cell proliferation and successful inhibition of the CXCL-CXCR2 pathway.

## 3. Discussion

PDAC is one of the most lethal cancer types, with patients having a median survival of 6 months and a five-year survival rate of only 11% [1,45]. Despite decades of effort, there has been no significant improvement in PDAC patient survival and, instead, PDAC is expected to become the second leading cause of cancer-related deaths by 2030 in the United States [46]. Characterized by a dense desmoplastic stroma of which CAFs are a predominant cellular component, PDAC therapy should leverage the potential for synergistic anti-tumor effects from CAF-targeted therapies. However, their multifaceted paradoxical role in tumor biology, as revealed by the failure of stromal and CAF depletion strategies, underscores the need for CAF-targeted drug delivery strategies that can aid in reprogramming the tumor microenvironment [9,28,47,48,49,50,51]. To this end, a comprehensive portfolio of PDAC CAF-selective proteins expressed on the cell surface is required. Current CAF markers include αSMA, S100A4, FAP, PDGFR-α/β, tenascin-C, CD90/Thy1, and PDPN. However, these molecules are not exclusively expressed by CAFs and can also be found in other cell types; such as in vascular muscle cells and pericytes (alpha-SMA), fibroblasts (S100A4), and CD45+ cells (FAP) [47,52,53]. The lack of selectivity to distinguish CAFs from normal cells limits their clinical use as molecular targets in targeted therapy. Here, we demonstrate the utility of an unbiased phage-display functional proteomic approach for the identification of a CAF-specific ligand (HTTIPKV) and its binding partner, TAS2R9, a previously uncharacterized CAF-selective marker. 

While bitter taste receptors (TAS2Rs) were canonically recognized as G-protein coupled receptors expressed in gustatory cells, studies have since demonstrated the expression of TAS2R family members across several extraoral tissues [54,55,56] and cancer types, including PDAC [57]. Gaida et al. found TAS2R38 localized with lipid droplets in pancreas-derived cancer cells and activation of TAS2R38-upregulated MAP kinases and a multidrug-resistance protein ABCB1 [58]. Stern et al. identified TAS2R10 expression in human PDAC tissue (79% cancer samples) and PDAC-derived cell lines, and demonstrated a TAS2R10-dependent signaling pathway that regulates ABCG2, a transmembrane drug-effluent pump that helps cells develop chemoresistance [59]. Interestingly, TAS2R9 and TAS2R10 are located adjacent to one another within chromosome 12. Overall, these studies highlight the functional role of bitter taste receptors in chemoresistance and immunosuppression of PDAC cancer cells. 

Bitter taste receptors have been demonstrated to mediate diverse physiologic roles [60]. While mounting evidence points to a role of bitter taste receptors in cancer epithelial cells [57] and in Tuft cells [61], this report, is to our knowledge, the first indication that TAS2R9 is expressed on CAFs. The fact that TAS2R9 is overexpressed in 7.33% of human PDAC tissue, and that its expression in pancreatic CAFs is 57-fold higher than in the human dermal fibroblast raises the interesting question of what role TAS2R9 may play in CAFs. In particular, the revelation that TAS2R9 mRNA expression is of prognostic significance in several cancers suggests its role could be tissue-specific or context-dependent. Overall, it remains to be elucidated what role TAS2R9 plays in tumor pathogenesis, whether that role may lead to new biological insight, and whether it could be targeted to improve patient outcomes. As a GPCR, TAS2R9 has strong potential as a drug target given that >50% of drugs currently on the market belong to this superfamily [60]. Furthermore, a pan-TAS2R agonist has demonstrated a favorable safety profile in a phase one clinical trial for metabolic and inflammation-related disease indications [62]; it was well tolerated at all dose levels up to 240 mg. Our work suggests that there may be a complex interplay between different taste receptors on multiple members of the greater tumor microenvironment. Future studies to elucidate the functional role of TAS2R9 in PDAC progression and its potential for novel therapeutic intervention need to be undertaken.

Here, we demonstrated that small molecule drugs encapsulated in TAS2R9-targeted nanoparticles enhanced the inhibition of tumor growth in a PDAC xenograft model. CXCR2 antagonists have been explored in non-cancer lung diseases (e.g., NCT01255592, NCT01006616) and in certain cancers such as castration-resistant prostate cancer (NCT03177187) and squamous cell carcinoma of the head and neck (NCT02499328). SB225002 is a small molecule CXCR2 inhibitor that has been shown to profoundly prolong survival in KPC mice [21]. Using this small molecule drug to test our targeting platform, we significantly decelerated tumor growth in mice treated with drugs in targeted liposomes compared to systematic delivery since day 12 post-treatment initiation. Successful delivery of SB225002 to interfere with CXCLs-CXCR2 response was confirmed with the reduced expression of CXCR2 compared to free-drug and untargeted SB225002 liposomes, respectively. Optimal CXCR2 suppression by CAF-targeted liposomes was also reflected in its superior constrained tumor growth and reduced cell proliferation. We, however, did not observe differences in tumor angiogenesis between treated and untreated mice. One potential explanation is that this is an effect of inhibition in other cell types or, alternatively, an effect of compensatory pathways [63]. Taken together, the CAF-targeted liposome improved the nanoparticle’s pharmacokinetics and enhanced the pharmacodynamic response in inducing antitumoral effects. 

In conclusion, we have identified TAS2R9 as a bitter receptor upregulated in pancreatic CAFs, adding to the growing body of research about extraoral expression of bitter taste receptors. Our work demonstrated the feasibility of using TAS2R9 as a molecular target to achieve stromal-targeting therapy and suggests that future studies should explore a pathogenic role of TAS2R9 in PDAC.

## 4. Materials and Methods

### 4.1. Animal Studies

All experiments were performed on male 6–8-week-old athymic nude mice purchased from Harlan Sprague Dawley Inc. (Indianapolis, IN, USA). All animal experiments were approved by the Animal Care and Use Committee at the University of Virginia (Protocol #3731). and conformed to the NIH “Guide for the Care and Use of Laboratory Animals in Research”.

### 4.2. Cell Culture and Reagents

CAF11-500 (Simeone Lab, NYU Langone Medical Center, New York, NY, USA), and BXCP3 (American Type Culture Collection, ATCC, Manassas, VA, USA) were grown in RPMI medium 1640 (Life Technologies, Carisbad, CA, USA). Human dermal fibroblasts (HDFs; Munson Lab, Virginia Polytechnic Institute and State University, Blacksburg, VA, USA) were grown in DMEM (Life Technologies). The RPMI and DMEM media were supplemented with 10% fetal bovine serum, 1% penicillin/streptomycin, and 1% L-glutamine. All cells were maintained in a humidified atmosphere containing 5% CO_2_ at 37 °C.

### 4.3. Lipids and Peptides for Liposome Preparation

1,2-Dioleoyl-sn-glycerol-3-phosphocholine (DOPC) and 1,2-distearoyl-sn-glycero-3- phosphoethanolamine-N-[methoxy(polyethylene glycol)-2000] (DSPE-PEG2000) were purchased from Avanti polar lipids, Miami, FL, USA; DSPE-PEG3400-maleimide was purchased from Laysan Bio Inc., Arab, AL, USA; 1,1′-Dioctadecyl-3,3,3′,3′-tetramethylindotricarbocyanine iodide (DiR) was purchased from Biotium Inc., Hayward, CA, USA; cholesterol was purchased from Millipore Sigma, Burlington, MA, USA. Peptides were synthesized by the Tufts University Peptide Synthesis Core Facility using standard FMOC chemistry and Rink-Amide resin (Tufts University, Boston, MA, USA).

### 4.4. Phage-Based Pulldown Assay

The binding partner of phage displaying HTTIPKV (phHTTIPKV) was identified using a phage display-based pulldown assay [64]. In brief, CAFs were cultured in 10 cm dishes overnight. 1 × 10^12^ plaque-forming unit (pfu) phHTTIPKV or control phage M13KE (200 μL, New England BioLabs, Ipswitch, MA, USA) were biotinylated in 5 μL of 0.2 μg/μL NHS-biotin in DMSO, 5 μL of 50 μg/μL sulfosuccinimidyl 2-[7-amino-4-methylcoumarin-3-acetamido]ethyl-1,3′dithiopropionate (sulfo-SAND) in DMSO, and 100 μL 50 mM carbonate buffer (pH 9.0) for 15 min, RT. Biotinylated phage were isolated by PEG/NaCl (2.5 mM NaCl + 80% *v*/*v* PEG-8000) precipitation and covalently cross-linked to CAFs by exposing them to 10 mW UV light for 15 min, on ice. Cells were lysed in 1 mL of PBS lysis buffer containing 1× protease inhibitor cocktail (Fisher Scientific, Hampton, NH, USA), 10 μL EDTA, and 10 μL Triton X-100. Cell lysates were then mixed with 200 μL Pierce Streptavidin Agarose (Thermo Scientific, Waltham, MA, USA) following the manufacturer’s instructions. The extracts were eluted in 50 μL of 50 mM NaCl/130 mM DTT for 15 min, and neutralized in 50 μL of 0.1 M glycine (pH 2.2) for 5 min. The eluted complexes were mixed with 4× Laemmli buffer (Bio-Rad, Hercules, CA, USA) and loaded into precast 4–15% tris-glycine eXtended (TGX) polyacrylamide gels (Bio-Rad, Hercules, CA, USA), followed by silver staining (SilverQuest Silver Staining Kit, Life Technologies, Carlsbad, CA, USA). Bands unique to phHTTIPKV were excised for mass spectrometry (MS/MS) analysis. The LC-MS system consisted of a Thermo Electron Orbitrap Velos ETD mass spectrometer system with a Protana nanospray ion source interfaced to a self-packed 8 cm × 75 um id Phenomenex Jupiter 10 um C18 reversed-phase capillary column. The peptides and proteins identified for the sample were displayed using Scaffold (v4.8.9) with the following settings (parent = 10 ppm, fragment = 1.00 Da, trypsin, 80% peptide threshold, 80% protein threshold).

### 4.5. TAS2R9 siRNA/shRNA Lentivirus Transduction

About 50,000 CAFs were cultured in a 24-well plate for 24 h, then treated with medium containing TAS2R9 (human) siRNA/shRNA lentivirus (piLenti-siRNA-GFP, Applied Biological Materials, Richmond, BC, Canada) at MOI = 10 with 8 μg/mL of polybrene. Fresh culture medium was replenished 24 h post induction of lentivirus. The transduced cells were sorted based on GFP expression using FACS Aria Fusion Cell Sorter (BD Biosciencee, San Jose, CA, USA) at the Flow Cytometry Core Facility at the University of Virginia. The sorted cells were cultured in medium containing 1 μg/mL puromycin (Millipore, Sigma, Burlington, MA, USA) to maintain stable cell lines.

### 4.6. Quantitative PCR

Cells were cultured in 10 cm dishes for 24 h, then RNA was isolated according to the manufacturer’s instructions (RNAeasy Mini kit, Qiagen, Hilden, Germany) and contaminating genomic DNA was removed with a DNase Max Kit (Qiagen). RNA concentration and purity were determined using a NanoDrop spectrophotometer (Thermo Fisher, Waltham, MA, USA). Complementary DNA (cDNA) was generated using QuantiTect Reverse Transcription Kit (Qiagen) and pre-amplified (Qiantitect SYBR Green PCR kit); 60 nM of each primer, TAS2R9 (Forward: 5′-GATGGTTCCCTTTATCCTTTGC-3′; Reverse: 5′-CCCTCATGTGGGCCTCTGTA-3′) and 18s (Forward: 5′-GTAACCCGTTGAACCCCATT-3′; Reverse: 5′-CCATCCAATCGGTAGTAGCG-3′) were mixed with cDNA template and SYBR green master mix and thermocycled as follows: 15 min at 95 °C, 55 cycles of [15 s at 94 °C, 30 s at 52 °C, and 45 s at 72 °C], then 4 min at 72 °C. The normalized gene expression was determined by the delta delta Ct method.

### 4.7. Phage ELISA

About 100 μL of 0.1 μg/mL rhTAS2R9 (Novus Biological, Littleton, CO, USA) was plated in 96-well plates overnight at 4 °C. Proteins were blocked with 100 μL of 5% milk for 30 min at room temperature, followed by two washes of 150 μL DPBS (Hyclone, Logan, UT, USA). The wild-type M13Ke phage and a known plectin-targeting phage, phKTLLPTP [26], were used as negative controls. For each phage, three wells were incubated with 40 μL of the HTTIPKV and the negative control phage (1 × 10^8^ pfu/μL) for 1 h at room temperature. Proteins were then washed three times with 150 μL DPBS, followed by incubation in HRP anti-M13 antibody (100 μL, 1:3000 dilution in 1% BSA/DPBS, Abcam) for 1 h at RT. Proteins were washed for four more times in 150 μL DPBS and 100 μL of TMB substrate solution (ThermoFisher Scientific, Waltham, MA, USA) was added. The absorbance was measured on a microplate reader at 650 nm at 35 min post addition of TMB

### 4.8. Western Blot Analysis

CAFs and TAS2R9 KD CAFs were cultured in 10 cm dishes until 95% confluency. Cells were washed with HBSS and lysed with radioimmunoprecipitation assay (RIPA) buffer (Thermo Scientific). The lysate was deglycosylated with PNGase F (New England BioLabs, Ipswich, MA, USA) following the manufacturer’s protocol. Lysate protein concentration was measured by the bicinchoninic acid assay (BCA, Pierce BCA Protein Assay Kit, Thermo Fisher Scientific, Waltham, MA, USA), and an equal amount of protein was loaded into the precast 4–15% TGX polyacrylamide gels (Invitrogen, MA, USA). The proteins were resolved by electrophoresis and transferred to nitrocellulose membranes. The membrane was incubated in denaturation buffer (62.5 mM Tris-HCl, pH 6.8, 2% SDS, and 100 mM βME) at 55 °C for 15 min. The membranes were then washed twice in PBS, blocked for 1 h in 50/50 TBS/Odyssey blocking buffer (LI-COR Bioscience, Lincoln, NE, USA), and incubated overnight at 4 °C with rabbit anti-TAS2R9 (Abcam; 1:100) and mouse anti-β-actin (Cell Signaling Technology, Danvers, MA, USA; 1:1000) in blocking buffer. The following day the membranes were washed and incubated with IRdye donkey anti-rabbit 680RD (Li-COR Bioscience; 1:5000) and IRdye donkey anti-mouse 800 CW (Li-COR Bioscience; 1:5000). Fluorescent signals were detected on the Li-COR Odyssey Fluorescent Imager analyzed with Image Studio Lite (LI-COR Bioscience, v5.2.5).

### 4.9. Flow Cytometry with TAS2R9 Primary Antibody

About 1 × 10^6^ trypsinized CAF cells were used for each condition: unstained, secondary antibody only, and TAS2R9 antibody (ThermoFisher Scientific, Waltham, MA, USA). Cell staining was conducted in Flow Cytometry Staining Buffer (eBioscience) for 1 h on ice. Cells were washed with flow buffer, then incubated with FITC goat anti-rabbit secondary antibody (Abcam) for 30 min on ice. After further washing, the samples were run on a BD Accuri C6 flow cytometer, and data were analyzed using FlowJo software v10.

### 4.10. Immunoflourscent Staining

About 40,000 cells were seeded in each well of a Millicell EZ SLIDE 4-well glass (Millipore, Burlington, MA, USA) and cultured for 24 h. Cells were washed with HBSS, fixed for 15 min with 4% PFA, washed in PBS, blocked for 1 h in 5% BSA in PBS, and then incubated for 1 h with anti-TAS2R9 antibody (Bioss Antibodies, Woburn, MA, USA) diluted in 1% BSA at 1:100. After further washing, cells were incubated for 1 h in goat anti-rabbit secondary antibody in 1% BSA in PBS (IgG H&L conjugated FITC, 1:200, Abcam, Cambridge, MA, USA). Then, the cells were washed and incubated with Wheat Germ Agglutinin Alexa Fluor 594 Conjugate at 5 μg/mL in PBS (Invitrogen) for 10 min. After a final set of washes, cells were mounted with ProLong Gold Antifade Mountant with DAPI (Thermo Fisher Scientific Inc., Waltham, MA, USA).

Tumor sections were blocked for 1 h in blocking buffer, and incubated with FITC-conjugated anti-CD31 (BD biosciences, 1:100, 20 min) then anti-rat secondary (1:250, 20 min), or anti-Ki67 (Abcam, 1:250, 20 min) then anti-rabbit secondary (1:250, 20 min). Images were collected using ZEISS LSM-880 Confocal Laser Scanning Microscope (Carl Zeiss Meditec, Inc., Jena, Germany). Mander’s correlation coefficients were determined using the JACoP plugin in ImageJ (National Institute of Health, Bethesda, MD, USA) for the colocalization analysis. To characterize CD31 and Ki67 expression, the area fraction of the positive pixels was measured for each image using the Measurement Tool in ImageJ. Section averages were entered into Prism to find overall means for each treatment group and tested for statistical significance between groups using the Student’s *t*-test.

### 4.11. Preperation and Characterization of Liposomes

[H]-HTTIPKVGGSK(fitc)C-[NH2] peptide was synthesized and purified at Tufts University Peptide Synthesis Core Facility. Drug-encapsulated liposomes were prepared using the thin-film hydration method [65]. In brief, 4 mg of FITC-labeled peptide was dissolved in 900 μL of degassed PBS/1 mM EDTA and 9 mg of DSPE-PEG3400-maleimide dissolved in 100 μL methanol. The two solutions were combined while bubbling with argon gas and then freeze-dried. About 20.5 mg DOPC, 9.5 mg DSPC-cholesterol, 9.5 mg DSPE-PEG2000, 1 mg freeze-dried DSPE-PEG3400- maleimide-conjugated peptide, and 0.5 mg DiR (Invitrogen) in methanol (25 mg/mL) were dissolved in 2 mL chloroform (Sigma-Aldrich, St. Louis, MO, USA). After evaporation, the lipid layer was hydrated by adding 2 mL saline and subjected to three freeze–thaw cycles. Drug-loaded liposomes were prepared with the same procedure but rehydrated in the solution containing SB225002 (Tocris Biosceince, Bristol, UK), 0.5 mg/mL instead of saline. The liposomes were sized by passing the solution 41 times through a manual extruder with a 0.2 μm Nuclepore filter (Thermo Fisher Scientific, Inc., Waltham, MA, USA). The size-extruded liposomes were characterized by Nanosight NS300 (Malvern Instruments Ltd., Worcestershire, UK) to determine the particle size and concentration. Zeta potential was measured using a ZetaSizer 3000 HSA (Malvern Panalytical, Westborough, MA, USA) in 10 mM HEPES buffer (pH 7.4) at 25 °C. The drug encapsulation was determined by Ultrospec 3000 UV/visible spectrophotometer (Pharmacia Biotech, Stockholm, Sweden).

### 4.12. Liposome Binding Assay

Biolayer interferometry (BLI) was performed using ForbeBio Octect Red 96 system (ForteBio, Menlo Park, CA, USA) in black 96-well plates (Nunc F96 Micro Well plates, Thermo Fisher). The total working volume for samples or buffer was 0.2 mL per well. Equilibration and loading steps were set to 1000 rpm; association and dissociation steps were carried out at 600 rpm. Prior to each assay, anti-HIS biosensor tops were pre-wetted in 0.2 mL PBS for at least 10 min, equilibrated with PBS for 100 s, then non-covalently loaded with his-tagged TAS2R9 (50–200 g/mL, 100 c). Subsequently, association with no peptide liposomes and HTTIPKV liposomes (40 μM) was carried out for 300 s. The dissociation was monitored in PBS for 600 s.

### 4.13. In Vivo Tumor Studies

For tumor implantation, BXPC3 cells at 500,000 cells/25 μL DPBS were combined with CAF cells at 1,500,000 cells/25 μL DPBS and 50 μL of Matrigel (BD Biosciences, San Jose, CA, USA) and injected subcutaneously into nude mice on the flanks, two tumors per animal. The tumor volume was calculated from the caliper measurements using the formula (width^2^ × length)/2.

For the liposome pharmacokinetic study, after tumors reached the size of 100 mm^3^, dye-labeled liposomes (50,000 pmol DiR) were injected intravenously via the tail vein. Fluorescence intensity was measured by fluorescence molecular tomography (FMT, PerkinElmer, Waltham, MA, USA) daily from day 0 to day 14 post liposome injection. Tumors were then harvested, submerged in Neg-50 Frozen Section Medium (Thermo Scientific), and snap frozen in liquid nitrogen vapor. Embedded tissues were cut into 5 μm sections using a cryostat (Leica Microsystems Inc., Buffalo Grove, IL, USA) for subsequent immunofluorescent staining.

For the SB225002 treatment study, after tumors reached the size of 100 mm^3^, mice were grouped into four treatment regimens (*n* = 5–6/group): (1) untreated, (2) free SB225002 (0.5 mg/kg per intraperitoneal injection, 5 times/week) [66], (3) SB225002-loaded untargeted liposomes (0.83 mg/kg per intravenous injection, 3 times/week), and (4) SB225002-loaded targeted liposomes (0.83 mg/kg per intravenous injection, 3 times/week). The drug dosage was calculated for each treatment to achieve the same weekly amount of Sb225002 given per animal. Tumor volume was measured with calipers twice weekly until day 19, at which time the tumors were harvested for cell proliferation (Ki67) and angiogenesis (CD31) analysis, and CXCR2 IHC staining.

### 4.14. Immunohistochemistry (IHC) Staining and Analysis

IHC staining for CXCR2 was performed on a robotic platform (Ventana discover Ultra Staining Module, Ventana Co., Tucson, AZ, USA). A heat-induced antigen retrieval protocol set for 64 min was carried out using a TRIS–ethylenediamine tetracetic acid (EDTA)–boric acid pH 8.4 buffer (Cell Conditioner 1; Roche Diagnostics, Indianapolis, IN, USA). Endogenous peroxidases were blocked with peroxidase inhibitor (CM1) for 8 min before incubating the cells with CXCR2 antibody (Abcam, Cat# ab 225732) at a 1:400 dilution for 60 min at room temperature. The antigen–antibody complex was then detected using the DISCOVERY anti-rabbit HQ HRP detection system and DISCOVERY ChromoMap DAB Kit (Roche Diagnostics, Indianapolis, IN, USA). All the slides were counterstained with hematoxylin, dehydrated, cleared, and mounted, then scanned using the Aperio ScanScope (Leica Biosystems, Buffalo Grove, IL, USA). Quantification of the percentage of DAB-positive cells was calculated as the ratio of positively stained cells to the total number of cells per tissue, irrespective of the localization using QuPath v.0.2.3 [44].

### 4.15. Pharmacokinetics

The accumulation coefficient (Ka) and clearance coefficient (Ke) for liposomes in tumors were determined using linear regression on log-transformed data from the two-week tumor accumulation time course. The fit line was evaluated and compared to the experimental data in MATLAB (The MathWorks Inc., Natick, MA, USA). Total liposome accumulation was estimated from the area under the curve from the projected fit line in MATLAB using a two-compartment model.

### 4.16. Statistical Analysis

Statistical analysis of the data was performed by student *t*-test. All in vitro data presented are expressed as mean ± standard deviation of at least three independent measurements. All in vivo data are plotted as mean ± standard error. For all comparisons, *p*-value < 0.05 was considered significant. All authors had access to the study data and reviewed and approved the final manuscript.

## Figures and Tables

**Figure 1 pharmaceuticals-16-00389-f001:**
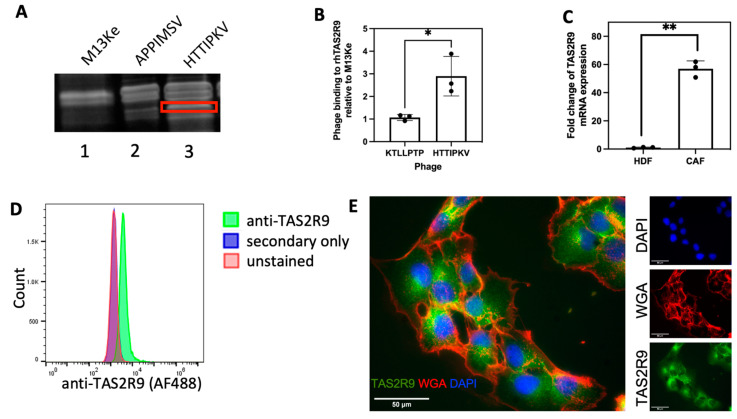
TAS2R9 expression in CAFs isolated from human PDAC. (**A**) Phage pulldowns from PDAC patient-derived CAFs by the M13Ke (lane 1) and CAF-specific phages, phAPPIMSV and phHTTIPKV (lane 2 and 3). A unique band (boxed) pulled-down by phHTTIPKV was sent for MS/MS analysis. (**B**) 4 × 10^9^ pfu phHTTIPKV and a PDAC specific phage, phKTLLPTP, were applied to rhTAS2R9 to determine the absorbance of phage binding to rhTAS2R9 at 650 nm. The absorbance of both phage were normalized to the absorbance detected from rhTAS2R9 bound to the wild-type M13Ke phage. *n* = 3, * *p* < 0.05. (**C**) qPCR analysis of TAS2R9 transcriptional levels in HDF and CAF cells. ** *p* = 0.009, *n* = 3. (**D**) Flow cytometry of anti-TAS2R9 demonstrates expression of TAS2R9 on the surface of CAF cells (representative graph of *n* = 3). (**E**) Immunofluorescence shows the distribution of TAS2R9 (green) relative to the membrane marker WGA (red) and nuclei (DAPI, blue), representative graph of *n* = 3 biological replicates.

**Figure 2 pharmaceuticals-16-00389-f002:**
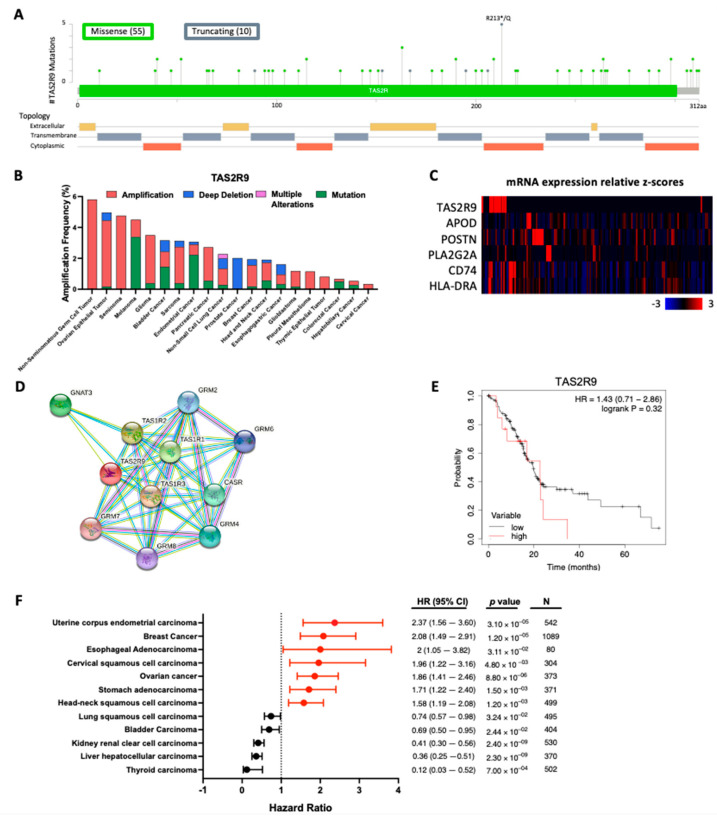
TAS2R9 mutation and mRNA analysis across several cancers. (**A**) Distribution of mutations and (**B**) alteration frequency of TAS2R9 based on the 32 TCGA pan-cancer atlas studies from cBioPortal [34,35]. The alteration frequency of datasets with >50 samples per cancer type is shown. (**C**) The mRNA expression of TAS2R9 and CAF subtype markers in PDAC RNA Seq samples from the PDAC TCGA PanCancer Atlas database (*n* = 150 (Appendix A [27]), APOD for iCAF, POSTN for myCAF, PLA2G2A for meCAF, CD74 for apCAF). (**D**) Human TAS2R9 proteins interactors were identified by querying the STRING database [36,37]. Network nodes represent interactor proteins, and colored lines represent the type of protein–protein association with a minimum interaction score of 0.400. (**E**) The Kaplan–Meier Plotter database was used to evaluate the prognostic utility of TAS2R9 mRNA expression in PDAC by pooling samples from 150 PDAC patients from the TCGA dataset [38]. (**F**) An expanded analysis of the prognostic values of TAS2R9 mRNA expression across different cancer types was assessed using RNAseq data from TCGA repositories through the Kaplan–Meier Plotter. The hazard ratios (HRs) and 95% confidence interval (CIs) are shown along with their associated log-rank *p* value and samples size (N). All publicly available datasets were accessed by (**A**–**C**) cBioPortal, (**D**) STRING, and (**E**,**F**) the Kaplan–Meier Plotter database on 25 August 2022.

**Figure 3 pharmaceuticals-16-00389-f003:**
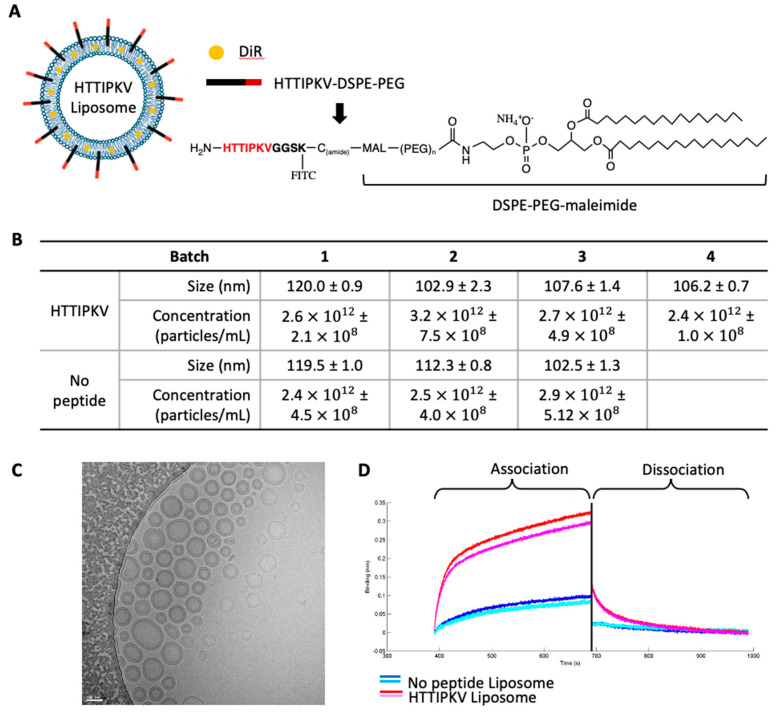
HTTIPKV-conjugated liposomes bind specifically to TAS2R9. (**A**) Schematics of HTTIPKV liposome. TAS2R9-targeting peptide (HTTIPKV) was conjugated to DSPE-PEG on DOPC liposomes. (**B**) Batch concentration and liposome size of targeted (HTTIPKV) and untargeted (no peptide) liposomes. (**C**) TEM of HTTIPKV liposomes, representative image of *n* = 6. (**D**) A binding assay using ForteBio showed the association and dissociation curves of liposomes with (red and pink) or without (dark and light blue) the HTTIPKV peptide to recombinant TAS2R9 protein.

**Figure 4 pharmaceuticals-16-00389-f004:**
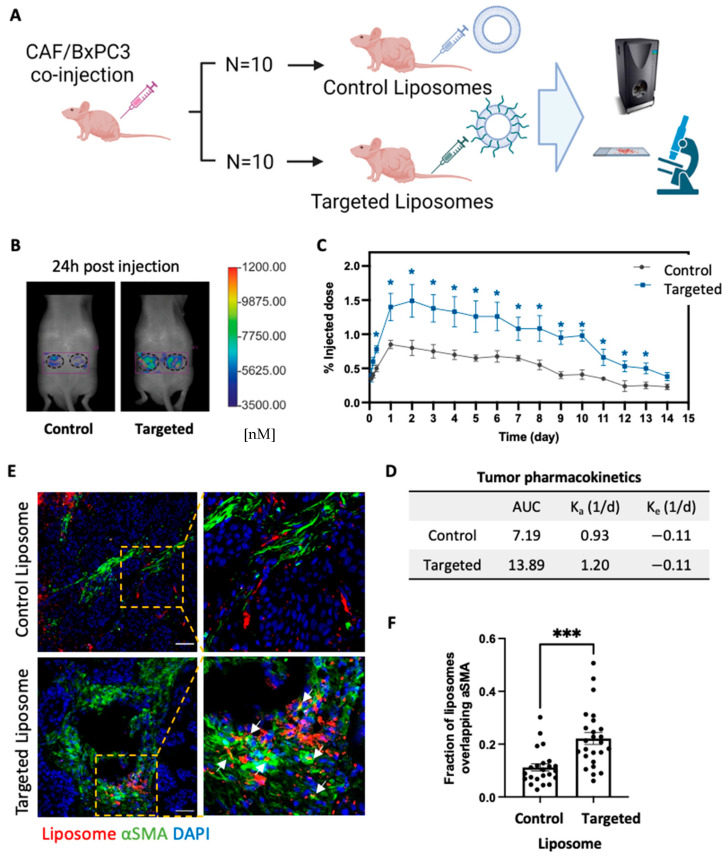
In vivo imaging of CAF-targeted liposomes. (**A**) Schematic of mouse study design. (**B**) FMT images of mice at 24 h post-liposome injection. (**C**) Mice bearing subcutaneous admix CAF/BXPC3 tumors (*n* = 10 tumors/group) were injected with dye-labeled liposomes, and the tumor accumulation was measured on an FMT using a region-of-interest around the tumor area. Statistical significance was measured with a Student’s *t*-test between targeted and control liposomes. * *p* < 0.05. (**D**) Tumor pharmacokinetics were determined by fitting the liposome time course data with compartment models by regression analysis in MATLAB. (**E**) Immunofluorescence images of tumor sections from mice injected with liposomes with and without HTTIPKV. The lipophilic dye shows the location of liposomes (red); cells were stained to show nuclei (DAPI, blue) and α-SMA-positive cells (green). Arrows indicate co-localization. Scale bars, 50 µm. (**F**) Mander’s colocalization analysis of injected liposomes overlapping with αSMA-positive cells in the admix tumor section at 24 h post liposome injection. *n* = 7~9 images/mouse, 3 mice/group. *** *p* = 0.0002.

**Figure 5 pharmaceuticals-16-00389-f005:**
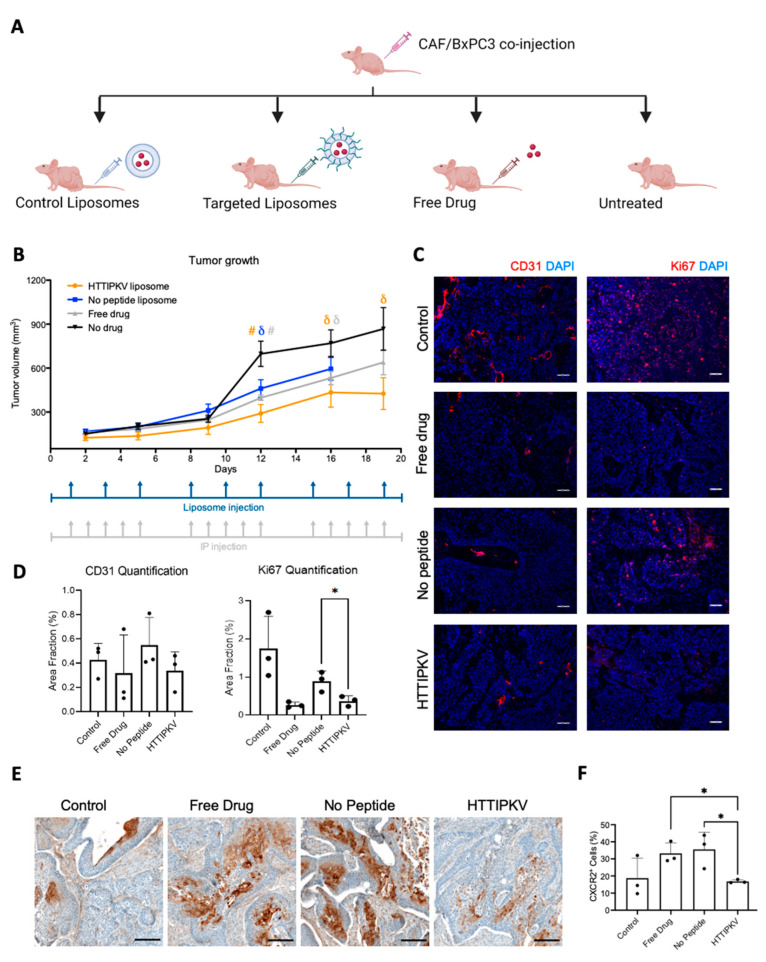
CXCR2 inhibition in an admix BXPC3/CAF xenograft model following SB225002 delivery. (**A**) Schematic of mouse study design. (**B**) Mice bearing subcutaneous admix BXPC3/CAF tumors were injected with SB225002, control liposomes loaded with SB225002, targeted liposomes loaded with SB225002, or no treatment. Tumor growth was measured via calipers for 19 days (*n* = 10–12 tumors/group). Dosage schedules for liposomes and free drug (IP injections) are as indicated. # *p* < 0.01, and δ *p* < 0.05 statistical significance between liposomes with or without a peptide compared to no drug by Student’s *t*-test. The untargeted liposome group was terminated early due to tumor ulceration. (**C**) Representative images of CD31 and Ki67 staining show the vessels and cell proliferation in different treatment groups. Scale bar, 50 µm. (**D**) Quantification of images using thresholding in ImageJ was averaged across five images of three sections of three tumors (45 images total per treatment group). * *p* < 0.05 statistical significance with Student’s *t*-test. (**E**) Representative images of anti-CXCR2 IHC staining. Scale bar, 150 µm. (**F**) Quantification of the percent of positively stained cells relative to the total number of cells per tissue, regardless of localization using QuPath v.0.2.3. (*n* = 3 tumors/group) [44]. * *p* < 0.05 statistical significance with Student’s *t*-test.

## Data Availability

Data is contained within the article and Appendix A.

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
