# Peer review of "A Bitter Taste Receptor as a Novel Molecular Target on Cancer-Associated Fibroblasts in Pancreatic Ductal Adenocarcinoma"

_pharmaceuticals, 2023, doi:10.3390/ph16030389_

Round 1
Reviewer 1 Report
The authors present an interesting study that describes the role of CAF markers in cancer progression and the use of CAF-specific ligands in PDAC treatment. The manuscript is well it is well thought out, and the results sound interesting, and it should be accepted for publication, but some minor revisions are required:
- A graphical abstract must be provided
- In the first lines of the introduction the authors discuss the unique desmoplastic stroma of PDAC and the presence of an intense fibro-inflammatory reaction, known as desmoplastic reaction (DR), which are involved in the cancer progression and resistance. To give more information, I suggest introducing the following recent review, along with references 3 and 4 (at line 42):
Cancers 2020, 12(11), 3206; https://doi.org/10.3390/cancers12113206
- In vitro and in vivo should be stated in italics
- LINE 330: Add a full stop at the end of the sentence
- LINE 437: CO2, 2 should be subscripted
- LINE 450: 1012, 12 should be superscripted
Reviewer 2 Report
Hung et al. identify a CAF restricted marker (TAS2R9) and its utility for targeted delivery of drugs in a pancreatic xenograft mouse model. The authors demonstrate the use of a CAF homing peptide to target liposomes to stromal heavy PDAC tumor microenvironment.
Major Critiques:
1. The main argument for using targeted liposomes over systemic treatment is to be able to reduce toxicities, which has not been evaluated in present study. Simple metrics like changes in body weight, liver enzymes, etc. should be included.
2. PDAC TCGA dataset does not include 177 PDAC tumor samples. Current analysis probably includes PNETs, normal, mets, etc. Please include details of samples included.
3. The authors use CTGF expression in CAFs as a surrogate marker for SB225002 efficacy. However, direct therapeutic efficacy of SB225002 loaded liposomes needs to be included to show that the treatment resulted in reduction of CXCR2 expression either through staining or flow cytometric analysis. This is especially important since Fig 5E does not show a clear decrease between Control and SB225002 liposome groups, even though the authors describe a decrease in CTGF expression (Fig 5F).
Minor Critiques:
1. What cell line was used for CAFs? Insufficient details in Materials Section
2. Figure S2B should say KD (Knockdown) and not KO (Knockout) since protein expression is not completely lost.
3. Figure S6 is missing
Round 2
Reviewer 2 Report
Authors have sufficiently addressed most concerns. Additionally, they have now provided detailed information about the TCGA dataset, however, it is not updated. Authors should refer to Table 1 in the following article and remove non-PDAC samples from their analysis.
Peran I, Madhavan S, Byers SW, McCoy MD. Curation of the Pancreatic Ductal Adenocarcinoma Subset of the Cancer Genome Atlas Is Essential for Accurate Conclusions about Survival-Related Molecular Mechanisms. Clin Cancer Res. 2018 Aug 15;24(16):3813-3819. doi: 10.1158/1078-0432.CCR-18-0290. Epub 2018 May 8. PMID: 29739787.
https://aacrjournals.org/clincancerres/article/24/16/3813/277807/Curation-of-the-Pancreatic-Ductal-Adenocarcinoma
